# Super-diffusion and crossover from diffusive to anomalous transport in a one-dimensional system

Anupam Kundu,

International Centre for Theoretical Sciences, Tata Institute of Fundamental Research, Bengaluru 560089, India

October 27, 2022

## Abstract

We study transport in a one-dimensional lattice system with two conserved quantities – 'volume' and energy. Considering a slowly evolving local equilibrium state that is slightly deviated from an underlying global equilibrium, we estimate the correction to the local equilibrium distribution. This correction arises mainly through the space-time correlations of the local currents. In the continuum limit, we show that the local equilibrium distribution along with the correction yields drift-diffusion equation for the 'volume' and super-diffusion equation for the energy in the linear response regime as macroscopic hydrodynamics. We find explicit expression of the super-diffusion equation. Further, we find diffusive correction to the super-diffusive evolution. Such a correction allows us to study a crossover from diffusive to anomalous transport. We demonstrate this crossover numerically through the spreading of an initially localized heat pulse in equilibrium as well as through the system size scaling of the stationary current in non-equilibrium steady state.

# 1  Introduction

In low dimensional systems, the transport of energy on macroscopic scale is often anomalous as manifested by emergence of super-diffusion [1–14]. According to the Green-Kubo formula the average current in a non-equilibrium system is related to the time integral of the equilibrium total current-current correlation in the linear response regime. Several numerical studies as well as theoretical arguments reveal that super-diffusion of energy is associated to the power-law tail of the current-current correlation at long time. The non-linear fluctuating hydrodynamic (NFHD) theory provides a general framework (applicable to a wide class of systems both Hamiltonian as well as stochastic) to understand this super-diffusion [4,5,15]. This theory describes the evolution of conserved fields on a mesoscopic scale in terms of hydrodynamic (HD) equations in which corresponding currents are expanded to non-linear order in the deviation from their values in a underlying global equilibrium (GE). For this, one assumes a slowly varying and slowly evolving local-equilibrium picture that is slightly deviated from a global equilibrium state of the system. Further the dissipation and the noise terms, obeying fluctuation-dissipation relation, are added phenomenologically to the currents in order to describe fluctuations. By decomposing the hydrodynamic evolution of the conserved fields into evolution of sound and heat modes (also known as normal modes), this theory reveals the connection between the super-diffusion (or anomalous transport) in translationally invariant Hamiltonian systems having short range interaction with the Kardar-Parisi-Zhang (KPZ) universality class [4,5,15,16]. This connection brought out by identifying the structural similarity between the stochastic HD equation of the sound mode fields with (coupled) noisy Burgers equations. However for the heat mode, which is non-propagating, one requires to study the sub-leading correction which is achieved through a mode-coupling approximation. The NFHD theory successfully applies to a wider class of Hamiltonian systems with short range interactions and the predictions of this theory classifies Hamiltonians into different universality classes depending on their transport behaviours.

In this paper, we study anomalous transport in a simple model defined on a one dimensional lattice of size $N$. Each lattice site contains a 'volume' variable $\eta_i$, that evolves according to

$$\dot{\eta}_i = V'(\eta_{i+1}) - V'(\eta_{i-1})$$
$$+ \text{ stochastic exchange } \eta \text{ between neighbouring sites at rate } \gamma \tag{1}$$

for $i = 1, 2, ..., N$, where $V(\eta) = \frac{k_o}{2}\eta^2$ with $k_o > 0$. We consider periodic as well as open boundary conditions in different situations, details of which will be provided in the particular sections. The stochastic exchange terms represent exchanging the variables across a bond at random with rate $\gamma$. This model was first introduced in [17] and was called "harmonic chain with volume exchange" (HCVE) system in [18]. The stochastic exchange terms are added to the dynamics to make the system posses good ergodic properties such that the system reach a thermal equilibrium state by itself. It is easy to see that this model has two globally conserved quantities, namely the total 'volume' $\sum_i \eta_i$ and total energy $\sum_i V(\eta_i)$ which yields two conservation equations for the corresponding locally conserved fields, again namely 'volume' density field and energy density field. Previously this model

was shown to exhibit anomalous energy transport in the non-equilibrium steady state (NESS) [15,17,18] and super-diffusion of space-time correlation in close system set-up (i.e. not connected to reservoirs) [10,15].

Following different microscopic approaches it was shown in [17] and [18], that the energy field (with the convective part subtracted) follows a super-diffusive evolution equation on the macroscopic scale. In the first part of the paper we provide a simpler alternative derivation of the macroscopic hydrodynamic equations corresponding to the conserved fields of the system in a close system set-up. We show that the non-convective part of the energy field evolves according to a super-diffusion equation and the 'volume' density field evolves diffusively in the linear response (LR) regime. We find explicit expressions of these equations. Additionally we find the diffusive correction to the super-diffusion equation explicitly. Our derivation is based on estimating the correction to the local-equilibrium distribution. Such a correction includes the contribution of current-current correlation to the computation of the average current which would appear in the macroscopic hydrodynamics. To compute these correlations, we invoke fluctuating hydrodynamics (FHD) on a mesoscopic scale. Our derivation is intuitive and reveals the importance and significance of the various approximations that goes into deriving the hydrodynamics on a macroscopic scale.

In the second part of the paper we demonstrate a crossover from diffusive transport to anomalous transport as one goes from mesoscopic length scale $\Lambda$ (*s.t.* $1 \ll \Lambda \ll N$) to macroscopic length scale. In NFHD, one can expect such a crossover from the structure of the the HD currents which have two parts: the Euler part that comes from the deviation from the global equilibrium and the second part constituting of dissipation (in the form of diffusion) and noise. The later part generically originates while coarse graining the conserved quantities, both in space and time. In the coarse graining procedure one usually replaces the values of the locally conserved fields by values averaged with respect to a local (canonical) thermal equilibrium distribution, which is different from the actual distribution of the microscopic degrees of freedom. The actual distribution involves correlations of the locally conserved fields among themselves as well as across space and time. The NFHD theory, in fact, computes contribution of these correlation to the steady state currents which could be anomalous in the leading order (of system size).

Although the actual values of the diffusion and noise terms in NFHD do not affect the leading anomalous behaviour of the energy current and similarly the super-diffusion, their presence is crucial for deriving the anomalous transport or super-diffusive behaviour. For a given Hamiltonian, deriving the fluctuating HD equations, especially the dissipation and the noise terms, is a difficult problem and until recently only a few attempts have been made. Using the projection operator technique and Markovian approximation, the study in [19] derives the diffusion and noise terms for a class of Hamiltonian systems defined on a one-dimensional lattice, although the expressions are not explicit. Another study that derives these terms involves one dimensional ideal gas of identical point particles undergoing stochastic collisions (both momentum and energy conserving) among three consecutive particles in addition to their Newtonian dynamics [20]. In the current study, we also provide a heuristic derivation of the dissipation and noise terms for the 'volume' field of the HCVE model in the LR regime. The presence of the stochastic exchange terms in Eq. (1) makes it possible to derive the noise terms explicitly.

While the diffusion terms do not affect the leading anomalous system size scaling of the steady state current in the LR regime, they can provide the sub-leading correction which has normal scaling as one finds in the Fourier's Law *i.e.* inversely proportional to the system size [20]. This suggests a crossover from diffusive to anomalous transport as one increases the system size from a mesoscopic scale to macroscopic scale in comparison

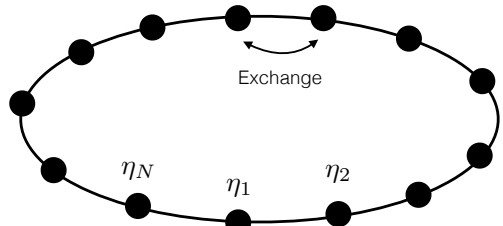

Figure 1: A schematic diagram of the HCVE model. Each site contains some variable $\eta_i$ which are called ''volume'. These variables are subjected to harmonic potential $V(\eta) = \frac{k_0 \eta^2}{2}$ at each lattice site. In addition to the deterministic evolution, 'volume' variables on successive sites are exchanged with rate $\gamma$.

to the microscopic scale (*e.g.* lattice spacing, interaction core or mean free path). In this paper we demonstrate such a crossover, both analytically and numerically through the system size scaling of the stationary current of the system in the non-equilibrium steady state (NESS) in the open system set-up.

The paper is organized as follows. In sec. 2, we describe the system and define the conserved quantities and, write the corresponding continuity equations. In the next section 3, we provide the derivation of the linearized hydrodynamic equations. This section starts with the Fokker-Planck equation and the solution of it. This solution is used to derive the hydrodynamic equations in two stages which are presented in sections 3.1 and 3.2. To complete the derivation of the linearized hydrodynamics as will be shown one requires to compute space-time correlation of the local currents, which is done in sec. 3.2.1. This section is followed by a demonstration of crossover from diffusive to anomalous transport in open system set-up in sec. 4. Finally in section 5, we provide a summary of our results along with possible future directions of study.

## 2 Conservation Laws and the equilibrium state

We first consider the HCVE model on a circular lattice of size $N$ with periodic boundary condition $\eta_{i+N} = \eta_i$ with $i = 1, 2, ..., N$. A schematic of the system is given in fig. 1. It is easy to see that the dynamics in Eq. (1) keeps the total volume and the total energy invariant *i.e.* $\frac{d}{dt} \sum_i \eta_i = 0$ and $\frac{d}{dt} \sum_i V(\eta_i) = 0$. The sum structure of these global conservations suggests the following two local conserved quantities, namely

$$\text{local 'volume': } \hat{h}_i(\vec{\eta}) = \eta_i,$$
$$\text{local 'energy': } \hat{e}_i(\vec{\eta}) = V(\eta_i) = \frac{k_o}{2} \eta_i^2, \tag{2}$$

where $\vec{\eta} = (\eta_1, \eta_2, ..., \eta_N)$. One can write the following conservation laws for these local quantities

$$\partial_t \hat{h}_i = \hat{j}_{i-1,i}^{(h)} - \hat{j}_{i,i+1}^{(h)}, \tag{3}$$
$$\partial_t \hat{e}_i = \hat{j}_{i-1,i}^{(e)} - \hat{j}_{i,i+1}^{(e)}, \tag{4}$$

where $\hat{j}_{i,i+1}^{(h)}$ and $\hat{j}_{i+1,i}^{(e)}$ are the local currents corresponding to the 'volume' field and the energy field, respectively. From the explicit form the dynamics given in Eq. (1) one finds

that the local currents have the following form

$$\hat{j}_{i,i+1}^{(h)}(\vec{\eta}, t) = -k_o(\hat{h}_i + \hat{h}_{i+1}) + (\hat{h}_i - \hat{h}_{i+1})(\gamma - \xi_{i+1/2}(t))$$
$$\hat{j}_{i,i+1}^{(e)}(\vec{\eta}, t) = -k_o^2(\hat{h}_i \hat{h}_{i+1}) + (\hat{e}_i - \hat{e}_{i+1})(\gamma - \xi_{i+1/2}(t)),$$
(5)

with $\xi_{i+1/2}(t) = \frac{dN_{i+1/2}(t)}{dt} - \gamma$ where $N_{i+1/2}(t)$ represents the Poisson process describing the exchanges happening at the bond $(i, i + 1)$ [represented by the subscript $(i + 1/2)$] with rate $\gamma$. It was shown in [17] that, for large $N$, the evolution given in Eq. (1) makes the system ergodic. Hence, using ensemble equivalence, one can describe the system by the following canonical ensemble distribution

$$P_{GE}(\{\eta_i\}) = \prod_{i=1}^{N} \sqrt{\frac{k_o}{2\pi T_0}} e^{-\frac{k_o}{2T_0}\left(\eta_i + \frac{\tau_0}{k_o}\right)^2},$$
(6)

where $T_0$ is the temperature and $\tau_0$ is the 'pressure' of the system. We have set the Boltzmann constant $k_B = 1$ throughout the paper. The quantities $T_0$ and $\tau_0$ are determined from the micro-canonical constraints for the total energy and total volume

$$\sum_{j=1}^{N} \langle \hat{h}_i(\vec{\eta}) \rangle_{P_{GE}} = N h_0, \quad \sum_{j=1}^{N} \langle \hat{e}_i(\vec{\eta}) \rangle_{P_{GE}} = N e_0,$$
(7)

with $h_0$ and $e_0$ being the 'volume' and the energy per particle, respectively which are constants both over space and time. In this paper $\langle ....\rangle_P$ denotes average with respect to a joint distribution $P$. In particular one finds the following equations for the equilibrium state

$$h_0 = -\frac{\tau_0}{k_o}, \qquad e_0 = \frac{T_0}{2} + \frac{\tau_0^2}{2k_o}.$$
(8)

# 3 Derivation of the hydrodynamic equations

Since the dynamics of the system is ergodic (in our case evolves to a homogeneous equilibrium state given by Eq. (6) at $t \to \infty$) and the microscopic currents in Eqs. (5) depend only on local variables, one may expect hydrodynamic evolutions to emerge for the locally conserved quantities, namely the 'volume' density and the energy density over coarse grained length and time scales. We assume a slowly evolving local equilibrium state for the system which is characterized by slowly varying conserved density fields at each time.

We start with the Fokker-Planck (FP) equation corresponding to the dynamics given in Eq. (1) which describes the evolution of the joint probability density $P(\vec{\eta}, t)$ of $\vec{\eta} = (\eta_1, \eta_2, ..., \eta_N)$ at time $t$. The FP equation is given by

$$\partial_t P(\vec{\eta}) = \mathcal{L}P(\vec{\eta}), \quad \text{with} \quad \mathcal{L} = \mathcal{L}_\ell + \mathcal{L}_{ex}.$$
(9)

Here $\mathcal{L}_\ell$ is the Liouvillian part and $\mathcal{L}_{ex}$ represents the contribution from the stochastic exchange events. Explicit expressions of these operators are given by

$$\mathcal{L}_\ell P(\vec{\eta}, t) = \sum_{i=1}^{N} \left[ V'(\eta_{i-1}) - V'(\eta_{i+1}) \right] \partial_{\eta_i} P(\vec{\eta}, t),$$
(10)

$$\mathcal{L}_{ex} P(\vec{\eta}, t) = \sum_{i=1}^{N} \left[ P(\vec{\eta}_{i,i+1}, t) - P(\vec{\eta}, t) \right],$$
(11)

where $\vec{\eta}_{i,i+1}$ represents the configuration after exchanging the $\eta$ variables at sites $i$ and $i+1$ and we impose periodic boundary condition. Note, for other boundary conditions the expressions of the FP operators will change. For these cases we will provide the expressions of the corresponding FP operators in the relevant sections later.

To solve the FP equation we follow a method similar to the one described in [21]. Starting from a LE state that is slightly deviated from the GE state, the solution of the FP equation (9) at a later time $t$ can be formally written as sum of the local equilibrium distribution $P_{LE}(\vec{\eta})$ plus a deviation $P_d(\vec{\eta}, t)$ from it

$$P(\vec{\eta}, t) = P_{LE}(\vec{\eta}, t) + P_d(\vec{\eta}, t). \tag{12}$$

The LE distribution $P_{LE}(\vec{\eta}, t)$, characterised by the local temperature field $T_i(t)$ and the 'pressure' field $\tau_i(t)$, is given by

$$P_{LE}(\vec{\eta}, t) = \prod_{i=1}^{N} \sqrt{\frac{k_o}{2\pi T_i(t)}} e^{-\frac{k_o}{2T_i(t)}\left(\eta_i + \frac{\tau_i(t)}{k_o}\right)^2}, \tag{13}$$

which implies the following local equations of state

$$h_i = \langle \hat{h}_i \rangle_{P_{LE}} = \langle \eta_i \rangle_{P_{LE}} = -\frac{\tau_i}{k_o},$$

$$e_i = \langle \hat{e}_i \rangle_{P_{LE}} = \langle V(\eta_i) \rangle_{P_{LE}} = \frac{T_i}{2} + \frac{\tau_i^2}{2k_o}. \tag{14}$$

The deviation $P_d(\vec{\eta}, t)$ from the LE distribution satisfies

$$\partial_t P_d(\vec{\eta}, t) - \mathcal{L} P_d(\vec{\eta}, t) = \mathcal{L} P_{LE}(\vec{\eta}, t) - \partial_t P_{LE}(\vec{\eta}, t). \tag{15}$$

A formal solution of this equation is given by

$$P_d(\vec{\eta}, t) = \int_{t_0}^{t} dt' e^{\mathcal{L}(t-t')} [\Phi(\vec{\eta}, t') - \Phi_{LE}(\vec{\eta}, t')] P_{LE}(\vec{\eta}, t'), \tag{16}$$

with $P_d(\vec{\eta}, t_0) = 0$ and

$$\Phi(\vec{\eta}, t) = P_{LE}(\vec{\eta}, t)^{-1} \mathcal{L} P_{LE}(\vec{\eta}, t) = \sum_{i=1}^{N} \left[ -\beta_0^2 \nabla_i T_i \hat{y}_{i,i+1}^{(e)} + \nabla_i \left(\frac{\tau_i}{T_i}\right) \hat{y}_{i,i+1}^{(h)} \right],$$

$$\Phi_{LE}(\vec{\eta}, t) = P_{LE}(\vec{\eta}, t)^{-1} \partial_t P_{LE}(\vec{\eta}, t) = \sum_{i=1}^{N} \left[ -\beta_0^2 (\partial_t T_i)_{LE} \; \hat{z}_i^{(e)} - \beta_0 (\partial_t \tau_i)_{LE} \; \hat{z}_i^{(h)} \right], \tag{17}$$

where $\nabla_i f_i = f_{i+1} - f_i$ represents the discrete forward difference and $\beta_0 = 1/T_0$. The expressions of $\hat{y}_{i,i+1}^{(h)}$, $\hat{y}_{i,i+1}^{(e)}$, $\hat{z}_i^{(h)}$ and $\hat{z}_i^{(e)}$ are given explicitly as

$$\hat{y}_{i,i+1}^{(h)}(\vec{\eta}) = -k_o(\hat{h}_i + \hat{h}_{i+1}) - \gamma(\hat{h}_i - \hat{h}_{i+1}), \tag{18}$$

$$\hat{y}_{i,i+1}^{(e)}(\vec{\eta}) = -k_o^2(\hat{h}_i \hat{h}_{i+1}) - \gamma(\hat{e}_i - \hat{e}_{i+1}), \tag{19}$$

$$\hat{z}_i^{(h)}(\vec{\eta}) = \left(\hat{h}_i + \frac{\tau_i}{k_o}\right), \tag{20}$$

$$\hat{z}_i^{(e)}(\vec{\eta}) = \frac{T_0}{2} - \frac{k_o}{2}\left(\hat{h}_i + \frac{\tau_i}{k_o}\right)^2, \tag{21}$$

where the functions $\hat{h}_i(\vec{\eta})$ and $\hat{e}_i(\vec{\eta})$ are provided in Eq. (2). To arrive at the expressions in Eq. (17) to Eq. (21), we have used the explicit form of $P_{LE}(\vec{\eta}, t)$ given in Eq. (13).

Moreover, while writing the explicit form of $\Phi_{LE}(\vec{\eta}, t)$ we have assumed the local deviation of the temperature and 'pressure' from their global equilibrium values $T_0$ and $\tau_0$ [see Eq. (8)] given, respectively, by $|T_i(t) - T_0| \ll T_0$ and $|\tau_i(t) - \tau_0| \ll |\tau_0|$ are small. The subscript $(...)_{LE}$ in the expression of $\Phi_{LE}$ indicates the rate of change of the $T_i(t)$ and $\tau_i(t)$ in local equilibrium state [shown later in Eq. (28)].

Note, the currents $\hat{y}_{i,i+1}^{(h)}$ and $\hat{y}_{i,i+1}^{(e)}$ are generated due to spatial inhomogeneity of the local temperature and 'pressure' fields in the LE state, whereas the the quantities $\hat{z}_i^{(h)}$ and $\hat{z}_i^{(e)}$ appearing due the time variations of these local fields. We will later see that the deviation from the LE, characterized by $P_d(\vec{\eta}, t)$ would incorporate the contributions from the space-time correlations of the local currents in the system.

The ansatz for the form of the joint distribution $P(\vec{\eta}, t)$ in Eq. (12) is sensible as a solution of the FP equation (9), when the deviations from the global equilibrium characterized by $\tilde{T}_i(t) = T_i(t) - T_0$ and $\tilde{\tau}_i(t) = \tau_i(t) - \tau_0$ are small *i.e.* the LE state is slightly deviated from the GE state. In this ansatz, we have introduced the space-time varying local temperature field $T_i(t)$ and the local 'pressure' field $\tau_i(t)$. Question is how should these fields evolve so that the ansatz for the $P(\vec{\eta}, t)$ in Eq. (12) holds to be a valid solution of the FP equation (9). The space-time evolution of these fields are determined by evaluating the continuity equations in (3) and (4) for the average 'volume' and energy fields. Performing average over the state $\vec{\eta}$ at time $t$ with respect to the joint distribution $P(\vec{\eta}, t)$ and average over the noises at the bonds $(i-1, i)$ and $(i, i+1)$ appearing from the exchange events, one finds

$$
\begin{aligned}
\partial_t h_i(t) &= j_{i-1,i}^{(h)}(t) - j_{i,i+1}^{(h)}(t), \\
\partial_t e_i(t) &= j_{i-1,i}^{(e)}(t) - j_{i,i+1}^{(e)}(t),
\end{aligned}
\tag{22}
$$

where the average currents are

$$
\begin{aligned}
j_{i,i+1}^{(h)}(t) &= \langle \hat{j}_{i,i+1}^{(h)}(\vec{\eta}, t) \rangle_P \\
j_{i,i+1}^{(e)}(t) &= \langle \hat{j}_{i,i+1}^{(e)}(\vec{\eta}, t) \rangle_P.
\end{aligned}
\tag{23}
$$

Our aim is to express these average currents in terms of the average fields $h_i(t) = \langle \hat{h}_i(\vec{\eta}) \rangle_P$, $e_i(t) = \langle \hat{e}_i(\vec{\eta}) \rangle_P$. Assuming the distribution $P$ in $\langle ... \rangle_P$ is given by Eq. (12) along with Eqs. (13) and (16), one can get evolution equations for the average conserved fields $h_i(t)$ and $e_i(t)$,— hence for the fields $\beta_i(t)$ and $\tau_i(t)$ through the equations of state in Eq. (14). We emphasize that we are interested to obtain linearized hydrodynamic evolutions of these fields. Our strategy consists of two steps. In the first step, we find the equations for the fields averaged only over the LE distribution in the LR regime. In this computation we will get a linearized HD equation which are diffusive in the form because one ignores the space-time correlations of the currents in this step (since the LE distribution in Eq. (13) is product in structure). Such correlations can provide extra contributions to the average currents at linear order in deviations from the GE state. To incorporate the effects of such correlations, we, in the second step, incorporate the contribution from the deviation $P_d(\vec{\eta}, t)$ from the LE distribution.

In order to get the proper space-time continuous hydrodynamic equations one uses the fact that the conserved fields vary slowly over space, *i.e.* their values change appreciably only over large number of lattice sites and the system size is very large *i.e.* $N \to \infty$. Equivalently, one can say that the temperature and 'pressure' fields vary slowly over the lattice as

$$
T_i(t) = T(\epsilon i, \bar{\epsilon} t) \text{ and } \tau_i(t) = \tau(\epsilon i, \bar{\epsilon} t),
\tag{24}
$$

where $\epsilon^{-1}$ and $\bar{\epsilon}^{-1}$ are macroscopic space and time scales measured in lattice (microscopic) units. In such situations one can formally replace the fields $e_i(t)$ and $h_i(t)$ by density functions $e(x,s)$ and $h(x,s)$, respectively where $x = i\epsilon$ and $s = t\bar{\epsilon}$. Similarly, the differences would get replaced by derivatives such as $\nabla_i f_i = f_{i+1} - f_i = \epsilon \partial_x f(x)$, $\Delta_i f_i = f_{i+1} - 2f_i + f_{i-1} = \epsilon^2 \partial_x^2 f(x)$ and the sums by integrals such as $\sum_i \rightarrow \int dx$. Also time derivative gets changed as $\partial_t f = \bar{\epsilon} \partial_s f$. We will use these continuum limits. However, we continue with the discrete notation for now and take the the continuum limits at appropriate stages.

Next we compute the average fields $h_i(t) = \langle \hat{h}_i(\vec{\eta}) \rangle_P$, $e_i(t) = \langle \hat{e}_i(\vec{\eta}) \rangle_P$ using the ansatz for $P(\vec{\eta}, t)$ in Eq. (12) and also evaluate the average currents $j_{i,i+1}^{(h)}(t)$ and $j_{i,i+1}^{(e)}(t)$ following the two steps mentioned above.

## 3.1 Contribution to the Linearized hydrodynamics from LE distribution

In order to compute the average fields $h_i(t) = \langle \hat{h}_i(\vec{\eta}) \rangle_P$, $e_i(t) = \langle \hat{e}_i(\vec{\eta}) \rangle_P$, we first make the approximation $P(\vec{\eta}, t) \approx P_{LE}(\vec{\eta}, t)$ assuming the deviations from the global equilibrium characterized by $\tilde{T}_i(t) = T_i(t) - T_0$ and $\tilde{\tau}_i(t) = \tau_i(t) - \tau_0$ are small. Keeping terms up to linear order in deviations, we get [from Eq. (14)]

$$h_i(t) \approx h_0 + \tilde{h}_i(t), \quad e_i(t) \approx e_0 + \tilde{e}_i(t), \quad \text{with}$$

$$\tilde{h}_i(t) = -\frac{\tilde{\tau}_i(t)}{k_o}, \quad \text{and} \quad \tilde{e}_i(t) = \frac{\tilde{T}_i(t)}{2} + \frac{\tau_o \tilde{\tau}_i(t)}{k_o}, \tag{25}$$

for the average values of the conserved fields and for the corresponding average currents, we get

$$j_{i,i+1}^{(h)}(t)\big|_{LE} \approx 2\tau_0 - k_o(\tilde{h}_i + \tilde{h}_{i+1}) + \gamma(\tilde{h}_i - \tilde{h}_{i+1})$$

$$j_{i,i+1}^{(e)}(t)\big|_{LE} \approx -\tau_0^2 - k_o^2 h_0(\tilde{h}_i + \tilde{h}_{i+1}) + \gamma(\tilde{e}_i - \tilde{e}_{i+1}). \tag{26}$$

where $h_0$ and $\tau_0$ are given in Eq. (8). Inserting these equations, on both sides of Eq. (22), and simplifying we get

$$\partial_t \tilde{h}_i(t) = k_o \nabla_i(\tilde{h}_i + \tilde{h}_{i-1}) + \gamma \Delta_i \tilde{h}_i,$$

$$\partial_t \tilde{e}_i(t) = k_o^2 h_0 \nabla_i(\tilde{h}_i + \tilde{h}_{i-1}) + \gamma \Delta_i \tilde{e}_i. \tag{27}$$

Using the equations (25), one can rewrite these equations in terms of $\tilde{T}_i(t)$ and $\tilde{\tau}_i(t)$ as

$$\partial_t \tilde{\tau}_i = k_o \nabla_i(\tilde{\tau}_i + \tilde{\tau}_{i-1}) + \gamma \Delta_i \tilde{\tau}_i$$

$$\partial_t \tilde{T}_i = \gamma \Delta_i \tilde{T}_i \tag{28}$$

The Eqs. (27) represent the linearized hydrodynamic equations when the space-time correlations among the currents are ignored. These equations can be improved by incorporating such correlations. In systems exhibiting normal transport, such correlations decay fast enough both in space and time that they effectively lead to (on macroscospic scale) diffusion of the locally conserved quantities. However, such correlations in our problem, as we will see, decay as power law (in time) in the leading order of (macroscopic) coarse graining scale leading to anomalous transport. In the next section we will see how these correlations appear to modify the equations (27).

## 3.2 Adding contribution to the linearized HD from the correction $P_d$ to $P_{LE}$

Recall that the equations (27) were obtained by computing averages of the locally conserved quantities with respect to $P_{LE}$. In this section we include the contribution from $P_d$ also to compute the average currents appearing in Eq. (22). The average currents for the conserved fields $u = (h, e)$, are computed as follows

$$
\begin{aligned}
j_{i,i+1}^{(u)}(t) &= \langle \hat{j}_{i,i+1}^{(u)}(t) \rangle_{P=P_{LE}+P_d} \\
&= \langle \hat{j}_{i,i+1}^{(u)}(t) \rangle_{P_{LE}} + \int_{t_0}^{t} dt' \hat{j}_{i,i+1}^{(u)}(\eta) \left\{ e^{\mathcal{L}(t-t')} [\Phi(\vec{\eta},t') - \Phi_{LE}(\vec{\eta},t')] P_{LE}(\vec{\eta},t') \right\}, \\
&= \langle \hat{j}_{i,i+1}^{(u)}(t) \rangle_{P_{LE}} + \int_{t_0}^{t} dt' \left\langle \hat{j}_{i,i+1}^{(u)}(t) [\Phi(t') - \Phi_{LE}(t')] \right\rangle_{P_{LE}}, \\
&= j_{i,i+1}^{(u)}(t) \big|_{LE} + \int_{t_0}^{t} dt' \left\langle \hat{j}_{i,i+1}^{(u)}(t) [\Phi(t') - \Phi_{LE}(t')] \right\rangle_{P_{GE}} + O(\tilde{T}^2, \tilde{\tau}^2, \tilde{T}\tilde{\tau}). \quad (29)
\end{aligned}
$$

While going from the third to fourth line we have changed the average $\langle ... \rangle_{P_{LE}}$ inside the integral to $\langle ... \rangle_{P_{GE}}$ because $[\Phi - \Phi_{LE}]$ is already in the linear order of the deviations from the GE characterised by $\tilde{T}_i$ and $\tilde{\tau}_i$. To see this more clearly, let us write $[\Phi - \Phi_{LE}]$ explicitly. First we recall $T_i(t) = T_0 + \tilde{T}_i(t)$ and $\tau_i(t) = \tau_0 + \tilde{\tau}_i(t)$. Using the forms of $(\partial_t \tilde{T}_i)_{LE}$ and $(\partial_t \tilde{\tau}_i)_{LE}$ from Eq. (28) in Eq. (17) and performing some manipulations one gets

$$
\begin{aligned}
\Phi(\vec{\eta},t) - \Phi_{LE}(\vec{\eta},t) = \sum_{i=1}^{N} & \Big[ -\beta_0^2 \nabla_i T_i(t) \left\{ \hat{y}_{i,i+1}^{(e)} + \tau_0 \hat{y}_{i,i+1}^{(h)} \right\} \\
& + \beta_0 \left\{ \nabla_i \tau_i(t) \hat{y}_{i,i+1}^{(h)} + k_o \nabla_i (\tau_i + \tau_{i-1}) \hat{z}_i^{(h)} \right\} \Big] \\
& + \beta_0 \gamma \sum_{i=1}^{N} \Big[ \beta_0 \Delta_i T_i \; \hat{z}_i^{(e)} + \Delta_i \tau_i \; \hat{z}_i^{(h)} \Big].
\end{aligned} \quad (30)
$$

Now using the definitions of the currents $\hat{y}^{(u)}$ and the quantities $\hat{z}^{(u)}$ from Eqs. (18) - (21) for $u = (h, e)$ we get

$$
\begin{aligned}
\Phi(\vec{\eta},t) - \Phi_{LE}(\vec{\eta},t) = -\sum_{i=1}^{N} & \Big[ \beta_0^2 \nabla_i T_i(t) \; (k_o^2 h_0^2 + \hat{\mathfrak{T}}_{i,i+1}^{(e)}) + \beta_0 \nabla_i \tau_i(t) (2 k_o h_0 + \hat{\mathfrak{T}}_{i,i+1}^{(h)}) \Big] \\
& + O(\Delta_i T, \Delta_i \tau, \Delta_i \hat{h}, \Delta_i \hat{e}), \quad \text{where,}
\end{aligned} \quad (31)
$$

$$
\hat{\mathfrak{T}}_{i,i+1}^{(e)} = -k_o^2 \hat{\tilde{h}}_i \hat{\tilde{h}}_{i+1}, \quad \hat{\mathfrak{T}}_{i,i+1}^{(h)} = -k_o (\hat{\tilde{h}}_{i+1} - \hat{\tilde{h}}_i),
$$

$$
\text{with,} \quad \hat{\tilde{h}}_i = \hat{h}_i - h_0. \quad (32)
$$

Note $\hat{\mathfrak{T}}_{i,i+1}^{(e)}$ depends non-linearly on the deviations $\hat{\tilde{h}}_i$. The $\Phi_{LE}$ term cancels the non-gradient type convective terms that depend linearly on the deviations. Hence, as will see later [see Eq. (36)], the additional contribution to the average currents arising from the deviation $P_d$ appears through the space-time correlations of the non-linear (in deviations of the fields from GE) parts of the currents.

Further note that $\hat{\mathfrak{T}}_{i,i+1}^{(h)}$ is of the form $\nabla_i \hat{\tilde{h}}_i$ and is accompanied with $\nabla_i \tau_i$. It will not contribute at the leading order to the average current in Eq. (29). Similarly, other terms of the same form in Eq. (31) can also be neglected. Moreover the, constant parts of the currents, like $k_o^2 h_0^2$ and $-2 k_o h_0$ in Eq. (31) will also not survive after averaging over the GE in Eq. (29). Hence the right hand side of the expression for the average current $j_{i,i+1}^{(u)}(t)$

in Eq. (29) simplifies a lot and we finally get

$$\langle \hat{j}_{i,i+1}^{(u)}(t)\rangle_P \approx \langle \hat{j}_{i,i+1}^{(u)}(t)\rangle_{P_{LE}} - \beta_0^2 \int_0^{t-t_0} dt' \sum_{\ell=1}^{N} \nabla_\ell T_\ell(t) \, \langle \hat{j}_{i,i+1}^{(u)}(t')\hat{\mathfrak{T}}_{\ell,\ell+1}^{(e)}(0)\rangle_{P_{GE}}, \qquad (33)$$

with $u = (h, e)$, where we have used time translational invariance in global equilibrium. Since we are interested in the evolution at macroscopic time scale, we assume $t - t_0$ is very large and approximate the time integral by performing integration over $(0, \infty)$. As a consequence we get

$$\langle \hat{j}_{i,i+1}^{(u)}(t)\rangle_P \approx \langle \hat{j}_{i,i+1}^{(u)}(t)\rangle_{P_{LE}} - \beta_0^2 \int_0^{\infty} dt' \sum_{\ell=1}^{N} \nabla_\ell T_\ell(t) \, \langle \hat{j}_{i,i+1}^{(u)}(t')\hat{\mathfrak{T}}_{\ell,\ell+1}^{(e)}(0)\rangle_{P_{GE}}. \qquad (34)$$

In the next section we will see that $\hat{\tilde{h}}_i(t)$ [see difinition in Eq. (32)] satisfies a linear fluctuating equation with white Gaussian noise. Additionally, in global equilibrium the fields $\hat{\tilde{h}}_i$ are independent and distributed according to Gaussian with zero mean. Hence average over any odd power of $\hat{\tilde{h}}_i$, even at different times are zero. Since $\langle \hat{j}_{i,i+1}^{(h)}(t')\hat{\mathfrak{T}}_{\ell,\ell+1}^{(e)}(0)\rangle_{P_{GE}}$ involves odd powers of $\hat{\tilde{h}}_i$, as can be seen the from the expressions of the currents in Eq. (5) and Eq. (31), it is zero. Once again ignoring contributions involving higher order derivatives of the fields, we get

$$\langle \hat{j}_{i,i+1}^{(h)}(t)\rangle_P = \langle \hat{j}_{i,i+1}^{(h)}(t)\rangle_{P_{LE}} + O(\tilde{u}^2, \Delta_i \tilde{u})$$

$$\langle \hat{j}_{i,i+1}^{(e)}(t)\rangle_P = \langle \hat{j}_{i,i+1}^{(e)}(t)\rangle_{P_{LE}} - \beta_0^2 \sum_{\ell=1}^{N} \nabla_\ell T_\ell(t) \, \mathcal{M}_{i,\ell} + O(\tilde{u}^2, \Delta_i \tilde{u}) \qquad (35)$$

$$\text{where} \qquad \mathcal{M}_{i,\ell} = \lim_{\tau \to \infty} \int_0^{\tau} dt' \langle \hat{\mathfrak{T}}_{i,i+1}^{(e)}(t')\hat{\mathfrak{T}}_{\ell,\ell+1}^{(e)}(0)\rangle_{P_{GE}},$$
$$\text{with } \hat{\mathfrak{T}}_{i,i+1}^{(e)}(t) = -k_o^2 \hat{\tilde{h}}_i(t)\hat{\tilde{h}}_{i+1}(t). \qquad (36)$$

Note $\mathcal{M}_{i,\ell}$ is similar to a transport coefficient. In order to compute this coefficient one needs to solve for the stochastic field $\hat{\tilde{h}}_i(\vec{\eta}) = \hat{h}_i(\vec{\eta}) - h_0$ which is evolving according to Eq. (1). We proceed to do that in the next section.

### 3.2.1 Fluctuating hydrodynamics of the 'volume' field:

We first note that the coefficient $\mathcal{M}_{i,\ell}$ should be independent of $i, \ell$ for a finite size $N$ of the system. This quantity is $i, \ell$ dependent only when $N \to \infty$ limit is taken before the $\tau \to \infty$ limit is taken. Hence, it is more appropriate to rewrite this coefficient as

$$\mathcal{M}_{i,\ell} = k_o^4 \lim_{\tau \to \infty} \lim_{N \to \infty} \int_0^{\tau} dt' \langle \hat{\mathfrak{T}}_{i,i+1}^{(e)}(t')\hat{\mathfrak{T}}_{\ell,\ell+1}^{(e)}(0)\rangle_{P_{GE}},$$
$$\text{where recall, } \hat{\mathfrak{T}}_{i,i+1}^{(e)}(t) = -k_o^2 \hat{\tilde{h}}_i(t)\hat{\tilde{h}}_{i+1}(t). \qquad (37)$$

The NFHD theory provides a general method to compute the space-time current-current correlation [22]. For the HCVE model it was shown in [15] that there is a sound mode and a heat mode corresponding to the two conserved quantities. For the particular choice of the harmonic potential $V(\eta) = \frac{k_o \eta^2}{2}$, one finds that the sound mode satisfies a drift-diffusion equation whereas the heat mode depends nonlinearly on the sound mode. It was argued that the traveling peak of the space-time correlations of sound mode satisfies diffusive scaling whereas the same for the heat mode is described by a $\frac{3}{2}$-Lévy scaling function. Such a scaling suggests super-diffusive contribution to the evolution from the

current-current correlation $\langle \hat{\mathcal{J}}_{i,i+1}^{(e)}(t')\hat{\mathcal{J}}_{\ell,\ell+1}^{(e)}(0)\rangle_{PGE}$ in Eq. (37). A brief discussion on the NFHD theory for the HCVE model is provided in Appendix A. In the next we provide a detail computation of the $\langle \hat{\mathcal{J}}_{i,i+1}^{(e)}(t')\hat{\mathcal{J}}_{\ell,\ell+1}^{(e)}(0)\rangle_{PGE}$.

Since we are interested to compute $\mathcal{M}_{i,\ell}$ in $N \to \infty$ limit, one convenient way is to start with the dynamics in Eq. (1) on infinite line. Furthermore, we are interested on the evolution of the local fields on coarse grained length and time scales for which one may consider the following effective dynamics for $\tilde{\hat{h}}_i = \hat{h}_i - h_0$ as

$$\partial_t \tilde{\hat{h}}_i = k_o \nabla_i (\tilde{\hat{h}}_i + \tilde{\hat{h}}_{i-1}) + \gamma \Delta_i \tilde{\hat{h}}_i + \nabla_i [\sqrt{\mathcal{B}}\xi_{i+1/2}(t)] \tag{38}$$

for $i = ..., -2, -1, 0, 1, 2, ...$ with the boundary conditions $\tilde{\hat{h}}_i(t) \to 0$ for $i \to \pm\infty$ at any $t$. The noise $\xi_{i+1/2}(t)$ appearing from the exchange events at the $(i, i+1)$ bond, is a white Gaussian noise of zero mean and unit variance. The strength of the noise is given by $\mathcal{B} = 2\gamma\left(\frac{T_0}{k_o} + h_0^2\right)$. A hueristic derivation of the above stochastic equation is given in Appendix B.

The formal solution of Eq. (38) can be written as

$$\tilde{\hat{h}}_i(t) = \sum_j G_{i,j}(t)\hat{h}_j(0) + \sqrt{\mathcal{B}} \int_0^t dt' \sum_j G_{i,j}(t-t')\nabla_j \xi_{j+1/2}(t'), \tag{39}$$

where $G_{i,j}(t)$ is the Green's function. Inserting this solution in Eq. (37) and using the fact $\langle \tilde{\hat{h}}_i(0)\tilde{\hat{h}}_j(0)\rangle_{PGE} = \frac{\delta_{i,j}}{\beta_0 k_o}$ [proved using Eq. (6) along with Eq. (2)], one gets

$$\mathcal{M}_{i,\ell} = \frac{k_o^2}{\beta_0^2} \int_0^\infty dt'[G_{i,\ell}(t')G_{i+1,\ell+1}(t') + G_{i,\ell+1}(t')G_{i+1,\ell}(t')], \tag{40}$$

where $\delta_{i,j}$ is the Kronecker delta. The Green's function $G_{i,j}(t)$ satisfies the following equation

$$\partial_t G_{i,j} = k_o \nabla_i (G_{i,j} + G_{i-1,j}) + \gamma \Delta_i G_{i,j} + \delta_{i,j}\delta(t). \tag{41}$$

Note that the above equation can be interpreted as the FP equation of a drifted random walker moving on an infinite lattice with velocity $\mu = -2k_o$ and diffusion constant $\mathcal{D} = \gamma$. For large $|i-j|$ and $t$, the Green's function has the following scaling form

$$G_{i,j}(t) \simeq \frac{1}{\sqrt{t}}\mathcal{G}\left(\frac{i-j+2k_o t}{\sqrt{t}}\right), \quad \text{where } \mathcal{G}(z) = \frac{1}{\sqrt{4\pi\gamma}}e^{-\frac{z^2}{4\gamma}}. \tag{42}$$

Using this form for the Green's function in Eq. (40) and simplifying one gets

$$\mathcal{M}_{i,\ell} = \frac{k_o^2}{2\pi\gamma\beta_0^2} \int_0^\infty dt' \frac{1}{t'}\mathcal{G}\left(\frac{i-\ell+2k_o t'}{\sqrt{t'}}\right)\mathcal{G}\left(\frac{i-\ell+2k_o t'}{\sqrt{t'}}\right). \tag{43}$$

$$= \frac{k_o^2}{\pi\gamma\beta_0^2}e^{-\frac{2(i-\ell)k_o}{\gamma}} K_0\left[\frac{2|i-\ell|k_o}{\gamma}\right], \tag{44}$$

where $K_0(z)$ is modified Bessel function of second kind of zeroth order. Inserting the expressions of the current in LE from Eq. (26) in Eq. (35), we get the following expressions of the average currents

$$\begin{aligned} j_{i,i+1}^{(h)}(t) &\approx 2\tau_0 - k_o(\tilde{h}_i + \tilde{h}_{i+1}) - \gamma\nabla_i\tilde{h}_i, \\ j_{i,i+1}^{(e)}(t) &\approx -\tau_0^2 - k_o^2 h_0(\tilde{h}_i + \tilde{h}_{i+1}) - \gamma\nabla_i\tilde{e}_i - \beta_0^2\sum_\ell \mathcal{M}_{i,\ell}\nabla_\ell T_\ell(t). \end{aligned} \tag{45}$$

Further inserting these expressions of the average currents in the continuity equations (22) we get

$$\partial_t \tilde{h}_i(t) = k_o \nabla_i(\tilde{h}_i + \tilde{h}_{i-1}) + \gamma \Delta_i \tilde{h}_i,$$
$$\partial_t \tilde{e}_i(t) = k_o^2 h_0 \nabla_i(\tilde{h}_i + \tilde{h}_{i-1}) + \gamma \Delta_i \tilde{e}_i + 2\beta_0^2 \nabla_i \sum_\ell \mathcal{M}_{i,\ell} \left( \tau_0 \nabla_\ell \tilde{h}_\ell + \nabla_\ell \tilde{e}_\ell \right), \tag{46}$$

where we have used the relation between $\tilde{T}_\ell$ with $\tilde{e}_\ell$ and $\tilde{h}_\ell$ from Eq. (25) and neglected terms involving higher order in deviations. Comparing these equations with Eqs. (27), we observe that the evolution equation for the 'pressure' field did not get modified while the equation for the energy field got modified after incorporating the contribution from the deviation $P_d$ from the local equilibrium distribution [see Eq. (12)]. In terms of the local 'pressure' deviation field $\tilde{\tau}_i = \tau_i - \tau_0$ and the local temperature deviation field $\tilde{T}_i = T_i - T_0$, these equations read

$$\partial_t \tilde{\tau}_i(t) = k_o \nabla_i(\tilde{\tau}_i + \tilde{\tau}_{i-1}) + \gamma \Delta_i \tilde{\tau}_i, \tag{47}$$

$$\partial_t \tilde{T}_i(t) = \gamma \Delta_i \tilde{T}_i + 2\beta_0^2 \nabla_i \sum_\ell \mathcal{M}_{i,\ell} \nabla_\ell \tilde{T}_\ell(t). \tag{48}$$

where we have used the equations of state in Eq. (25).

## 3.3 Continuum limit

We take continuum limit as discussed around Eq. (24) by making the transformation $i \to x = i\epsilon$ and $t \to s = \bar{\epsilon}t$. Let us first focus on the 'pressure' field equation (47). Observe that the field $\tilde{\tau}_i(t)$ has a ballistic propagation with velocity $\mu = -2k_o$. This suggests us to expect the following scaling form $\tilde{\tau}_i(t) \to \tilde{\mathcal{T}}((i + 2k_o t)\epsilon, \bar{\epsilon}t)$ for the pressure field. This scaling density function evolves according to

$$\partial_s \tilde{\mathcal{T}}(z,s) = \frac{\epsilon^2}{\bar{\epsilon}} \gamma \partial_x^2 \tilde{\mathcal{T}}(z,s), \quad \text{with} \quad z = (i + 2k_o t)\epsilon. \tag{49}$$

Choosing diffusive space-time scaling $\bar{\epsilon} \sim \epsilon^2$, we find that the 'pressure' field moves ballistically and spreads diffusively at the ballistic front.

For the temperature field, we consider the scaling form $\tilde{T}_i(t) \to \tilde{T}(i\epsilon, \bar{\epsilon}t)$ and get,

$$\partial_s \tilde{T}(x,s) = \frac{\epsilon^2}{\bar{\epsilon}} \gamma \partial_x^2 \tilde{T}(x,s) + \frac{\epsilon^{3/2}}{\bar{\epsilon}} \frac{k_o^{3/2}}{\sqrt{\pi\gamma}} \partial_x \int dy \frac{\Theta(y-x)}{\sqrt{y-x}} \partial_y \tilde{T}(y,s). \tag{50}$$

To arrive at the above equation we have used the below limit

$$\lim_{\epsilon \to 0} \epsilon^{-1/2} 2\beta_0^2 \mathcal{M}_{i=x/\epsilon, \ell=y/\epsilon} \to \frac{k_o^{3/2}}{\sqrt{\pi\gamma}} \frac{\Theta(y-x)}{\sqrt{y-x}}, \tag{51}$$

where $\Theta(x)$ is Heaviside theta function. From this equation one can get different hydrodynamic evolutions, depending on the choice of the space-time scaling for the coarse graining.

- *Ballistic space-time scaling i.e. $\bar{\epsilon} = \epsilon$*: In this case one finds

$$\partial_s \tilde{T}(x,s) = \epsilon\gamma \partial_x^2 \tilde{T}(x,s) + \sqrt{\epsilon} \frac{k_o^{3/2}}{\sqrt{\pi\gamma}} \partial_x \int dy \frac{\Theta(y-x)}{\sqrt{y-x}} \partial_y \tilde{T}(y,s) \approx 0 \tag{52}$$

which indicates that the temperature profile does not evolve.

- *Super-diffusive scaling i.e.* $\bar{\epsilon} = \epsilon^{3/2}$: For this case one finds

$$\partial_s \tilde{T}(x,s) = \frac{k_o^{3/2}}{\sqrt{\pi\gamma}} \left[ \partial_x \int dy \frac{\Theta(y-x)}{\sqrt{y-x}} \partial_y \tilde{T}(y,s) + \sqrt{\frac{\epsilon}{\epsilon_c}} \partial_x^2 \tilde{T}(x,s) \right], \qquad (53)$$

$$\text{with} \quad \epsilon_c = \pi \left( \frac{k_o}{\gamma} \right)^3. \qquad (54)$$

This equation implies that the local temperature field (equivalently the energy density field) performs super-diffusion at large length and time scales. The diffusive correction suggests a crossover from diffusive evolution at shorter space-time scale $(x\sqrt{s})$ to super-diffusive evolution on larger space-time scale $(x \sim s^{2/3})$. The crossover occurs at a length scale $N_c \sim \epsilon_c^{-1}$. This means, if one observes the evolution of an initially localized pulse, then at shorter time scale the pulse will spread diffusively as long as the spread is smaller than $N_c$. But at larger times when the amount of spread becomes larger than $N_c$, the spreading happens super-diffusively. It seems harder to see this crossover numerically with time. Instead we study the spreading of an initially localized pulse in two cases, very large $N_c$ and very small $N_c$, for which we should observe diffusive and super-diffusive spreading respectively. We demonstrate this in fig. 2 where we plot the space-time scaling of the spreading of an initially localized temperature pulse. We consider a temperature pulse initially localized around $i = N/2$ on a periodic ring by choosing the initial configuration $\vec{\eta}(0)$ from the distribution given in Eq. (13) with $\tau_i(0) = 0$ and $T_i(0) = 1.0 + 0.5\, g_i$ where $g_i$ is a Gaussian function of $i$, centered at $i = N/2$ with variance 1.5. We consider two values of the harmonic strength $k_o = 0.02$ and $k_o = 1.0$. The corresponding crossover length scales are of order $N_c \sim 39788$ and $N_c \sim 1$, respectively. In fig. 2a we observe a diffusive scaling whereas in fig. 2b we observe a super-diffusive scaling as expected. The different scaling behavior in the two cases imply a crossover with time for a fixed set of parameters on a given size of ring that is quite large. Another way to demonstrate this crossover is to look at the system size scaling of the stationary current in NESS of the system in the open system set-up which we discuss in the next section.

## 4 Study in open system set-up and crossover from diffusive to anomalous transport

In this section we consider the open set-up in which we attach two Langevin reservoirs of different temperatures $T_L$ and $T_R$ at the two ends of the system. The dynamics in Eq. (1) is modified to

$$\begin{aligned}
\dot{\eta}_i = \;& V'(\eta_{i+1}) - V'(\eta_{i-1}) \\
& + \text{stochastic exchange } \eta \text{ between neighbouring sites at rate } \gamma \\
& + \delta_{i,1} \left( -\lambda V'(\eta_1) + \sqrt{2\lambda T_L}\, \zeta_L(t) \right) \\
& + \delta_{i,L} \left( -\lambda V'(\eta_L) + \sqrt{2\lambda T_R}\, \zeta_R(t) \right)
\end{aligned} \qquad (55)$$

with fixed boundary conditions $\eta_0 = \eta_{N+1} = 0$. Here $\zeta_{L,R}(t)$ are mean zero and unit variance white Gaussian noises and $\lambda$ is the strength of the dissipation into the bath (which we assume to be a constant of $O(1)$.). The FP equation now reads

$$\partial_t P(\vec{\eta}) = \mathcal{L} P(\vec{\eta}), \quad \text{with} \quad \mathcal{L} = \mathcal{L}_\ell + \mathcal{L}_{ex} + \mathcal{L}_b. \qquad (56)$$

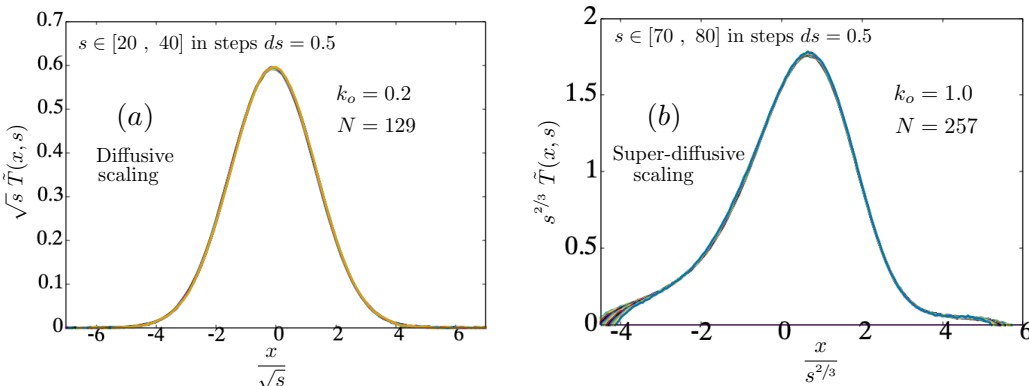

Figure 2: Space-time scaling of a initially localized temperature field for two cases (a) $k_o = 0.02$ and (b) $k_o = 1.0$. For the former case $N_c \sim 34888$ and for the later case $N_c \sim 1$. We observe diffusive scaling in (a) and super-diffusive scaling in (b) respectively.. The lines in the plot are obtained by numerically integrating the Langevin equations (1) on a periodic ring of size $N$ with time steps $dt = 0.005$ and averaged over $10^7$ realizations. Each curve corresponds to the (scaled) temperature profile at time $s$ chosen within a range after interval $ds = 0.5$. The ranges for the two plots are given in the respective legends.

The Liouvillian part $\mathcal{L}_\ell$ is given in Eq. (10) and for the stochastic exchange part $\mathcal{L}_{ex}$ in Eq. (11), the summation now runs from $i = 1$ to $(N-1)$. The boundary part $\mathcal{L}_b$ is given by

$$\mathcal{L}_b P(\vec{\eta}, t) = \lambda T_L \partial_{\eta_1}^2 P + \lambda \partial_{\eta_1} V'(\eta_1) P + \lambda T_R \partial_{\eta_N}^2 P + \lambda \partial_{\eta_N} V'(\eta_N) P. \tag{57}$$

For $T_L = T_R = T_0$ the dynamics in Eq. (55) takes the system to the global equilibrium state described by

$$P_{GE}(\{\eta_i\}) = \prod_{i=1}^{N} \sqrt{\frac{k_o}{2\pi T_0}} e^{-\frac{k_o}{2T_0}\eta_i^2}, \tag{58}$$

which implies $e_0 = \langle \hat{e}_i \rangle_{P_{GE}} = T_0/2$ and $h_0 = \langle \hat{h}_i \rangle_{P_{GE}} = 0, \ \forall i$.

When $T_L \neq T_R$, we write approximate solution of the FP equation as sum of local equilibrium distribution $P_{LE}$ plus a deviation from it as in Eqs. (12) to (16). We further assume $\delta T = T_L - T_R$ is small, hence the local equilibrium is slightly deviated from an underlying GE described by the distribution $P_{GE}$ in Eq. (58) with $T_0 = \frac{T_L + T_R}{2}$. The expression of $\Phi_{LE}(\vec{\eta}, t)$ remains same as in Eq. (17). However, the expression of $\Phi(\vec{\eta}, t)$ gets slightly modified due the presence of boundary currents from the baths and it now reads as

$$
\begin{aligned}
\Phi(\vec{\eta}, t) = &\sum_{i=1}^{N-1} \left[ -\beta_0^2 \nabla_i T_i \hat{y}_{i,i+1}^{(e)} + \nabla_i \left( \frac{\tau_i}{T_i} \right) \hat{y}_{i,i+1}^{(h)} \right] \\
&+ \lambda k_o \left[ \frac{\beta_1 - \beta_L}{\beta_L} \left( k_o \beta_1 \left( \hat{h}_1 + \frac{\tau_1}{k_o} \right)^2 - 1 \right) + \beta_1 \tau_1 \left( \hat{h}_1 + \frac{\tau_1}{k_o} \right) \right], \\
&+ \lambda k_o \left[ \frac{\beta_N - \beta_R}{\beta_R} \left( k_o \beta_N \left( \hat{h}_N + \frac{\tau_N}{k_o} \right)^2 - 1 \right) + \beta_N \tau_N \left( \hat{h}_N + \frac{\tau_N}{k_o} \right) \right],
\end{aligned} \tag{59}
$$

where $\beta_L = 1/T_L$ and $\beta_R = 1/T_R$.

Following the steps as done in sections 3.1, 3.2 and 3.2.1, one arrives at the same equations as in Eq. (46):

$$\partial_t \tilde{h}_i(t) = k_o \nabla_i (\tilde{h}_i + \tilde{h}_{i-1}) + \gamma \Delta_i \tilde{h}_i, \tag{60}$$

$$\partial_t \tilde{e}_i(t) = \gamma \Delta_i \tilde{e}_i + 2\beta_0^2 \nabla_i \sum_\ell \mathcal{M}_{i,\ell} \ \nabla_\ell \tilde{e}_\ell, \tag{61}$$

with the kernel $\mathcal{M}_{i,\ell}$ given in Eq. (40) and $\beta_0 = T_0^{-1} = \frac{2}{T_L + T_R}$ except for different boundary conditions. Note, unlike Eq. (46) there are no terms depending on the 'volume' field in Eq. (61). This is because $h_0 = \langle \hat{h} \rangle_{GE} = 0$ (hence $\tau_0 = 0$) in the GE as can be seen from the GE distribution in Eq. (58). The boundary currents in the expression of $\Phi(\vec{\eta}, t)$ in Eq. (59) do not contribute at the leading order in system size while calculating the average current. The main difference with the previous case is that the boundary conditions now are

$$h_{i=0} = 0, \ h_{i=N+1} = 0, \tag{62}$$

$$e_{i=0} = e_L = \frac{T_L}{2}, \ e_{i=N+1} = \frac{T_R}{2}, \tag{63}$$

which imply $\tilde{h}_{i=0} = 0$, $\tilde{h}_{i=N+1} = 0$ and $\tilde{e}_{i=0} = \frac{T_L - T_R}{4}$, $\tilde{e}_{i=N+1} = \frac{T_R - T_L}{4}$. To evaluate the kernel $\mathcal{M}_{i,\ell}$ given in Eq. (40), one needs to solve the Green's function equation (41) with absorbing boundary conditions $G_{i,j} = 0$ for $i$ or $j$ equal to 0 and $N+1$. In the scaling limit (as in Eq. (42), the Green's function is given by

$$G_{i,j}(t) = \frac{e^{-\frac{k_o(i-j)}{2\gamma t}} e^{-\frac{2k_o^2 t}{2\gamma}}}{\sqrt{4\pi\gamma t}} \sum_{p=-\infty}^{\infty} \left[ e^{-\frac{(i-j+2pN)^2}{4\gamma t}} - e^{-\frac{(i+j+2pN)^2}{4\gamma t}} \right], \tag{64}$$

using which in Eq. (40) one can show

$$\lim_{N \to \infty} \sqrt{N} 2\beta_0^2 \ \mathcal{M}_{i=xN, \ell=yN} \to \frac{k_o^{3/2}}{\sqrt{\pi\gamma}} \frac{\Theta(y-x)}{\sqrt{y-x}}. \tag{65}$$

Note in the continuum limit, we get the same position space representation of the kernel as in the infinite chain case studied in the previous section, however with different boundary conditions. A similar kernel was obtained for the harmonic chain with momentum exchange model for different boundary conditions [8, 23].

To take continuum limit as discussed in sec. 3.3 once again we make the transformation $i \to x = i\epsilon$ and $t \to s = \bar{\epsilon} t$. For the 'pressure' field we choose $\epsilon = N^{-1}$ and $\bar{\epsilon} = N^{-2}$ since one expects diffusive behavior for $\tilde{\tau}_i(t) \to \tilde{\mathcal{T}}((i + 2k_o t)\epsilon, \bar{\epsilon} t)$. We get drift-diffusion for $\tilde{\mathcal{T}}(z, s)$ as given in Eqs. (49). For the evolution of the temperature field, we again expect super-diffusive evolution for $\tilde{T}_i(t) \to \tilde{T}(i\epsilon, \bar{\epsilon} t)$ with $\epsilon = N^{-1}$, $\bar{\epsilon} = N^{-3/2}$ and we get same super-diffusive evolution as given in Eq. (53).

The boundary conditions in Eq. (62) implies that the 'volume' profile decays to zero everywhere in the non-equilibrium steady state (NESS). On the other hand, the temperature profile $T_{ss}(x) = T_0 + \delta T \Psi(x)$ in the steady state satisfies

$$\frac{k_o^{3/2}}{\sqrt{\pi\gamma}} \partial_x \int_0^1 dy \frac{\Theta(y-x)}{\sqrt{y-x}} \partial_y \Psi(y) + \sqrt{\frac{1}{N}} \gamma \partial_x^2 \Psi(x) = 0, \ \text{for} \ 0 \le x \le 1, \tag{66}$$

with $\Psi(0) = 1/2$ and $\Psi(1) = -1/2$. The energy current in the steady state can be read off from Eq. (45) with $h_0 = 0$ and is given by $J_{ss} = -\gamma \nabla_i e_i + 2\beta_0^2 \sum_\ell \mathcal{M}_{i,\ell} \nabla_\ell e_\ell$

which in the continuum limit can be expressed in terms of the temperature profile using $2e(x) = T_{ss}(x) = T_0 + \delta T \Psi(x)$. It reads as

$$J_{ss} = -\frac{1}{N}\frac{\gamma}{2}\partial_x \Psi_{ss}(x) - \frac{1}{\sqrt{N}}\frac{k_o^{3/2}}{2\sqrt{\pi\gamma}}\int_0^1 dy\,\frac{\Theta(y-x)}{\sqrt{y-x}}\partial_y \Psi_{ss}(y). \tag{67}$$

The equation (66) ensures that the stationary current $J_{ss}$ is $x$ independent. Hence integrating both sides of Eq. (68) with respect to $x$ over $[0,1]$, one gets

$$J_{ss} = \frac{\delta T}{2}\left[\frac{1}{\sqrt{N}}\frac{k_o^{3/2}C}{\sqrt{\pi\gamma}} + \frac{1}{N}\gamma\right], \quad \text{with} \quad C = \int_0^1 dx \int_0^1 dy\,\frac{\Theta(y-x)}{\sqrt{y-x}}\partial_y \Psi(y). \tag{68}$$

At this stage, few comments are in order.

– *Anomalous scaling*: From Eq. (68) we see that the stationary current decays anomalously with as $\sim \frac{1}{\sqrt{N}}$ at the leading order in $N$. This anomalous scaling was obtained previously in [17] and [18] using methods different from the one presented here. The $O(1/N)$ term provides a diffusive correction to the anomalous scaling.

– *Non-local Fourier's law:* The expression of the current in Eq. (68) is a non-local linear response relation, which is drastically different from the usual Fourier's law. In local Fourier's law, the current at any point $x$ in the system is directly proportional to the local derivative of the temperature profile. On the other hand, in the non-local version, the current at any point $x$ gets contribution from the derivative of the temperature profile at other points also. Such non-local generalisation of Fourier's law was also obtained in few other systems [7–9, 23–25] and it implies a non-local generalization of the heat diffusion equation as we have obtained in Eq. (53). Such a generalization of the heat diffusion equation was obtained for the HCVE model in [18] by computing the microscopic two-point correlations $\langle \eta_i \eta_j \rangle_{P_{ss}}$ in the steady state described by a stationary distribution $P_{ss}(\vec{\eta})$. In this paper we have re-derived the same equation using a different method along with a diffusive correction.

– *Nonlinear temperature profile $\Psi(x)$:* Neglecting the diffusive part one can solve the non-local part of Eq. (66) and the solution is given by $\Psi(x) = \sqrt{1-x} - 1/2$ [18]. Using this solution in Eq. (68) one finds $C = \frac{\pi}{2}$. Unlike the (purely) diffusive case, the temperature profile $\Psi(x)$ is non-linear and singular (has diverging derivative at the right boundary). Similar non-linear and singular temperature profiles were also obtained in different contexts, such as in momentum exchange model [8, 23, 24], in hard-point gas [26, 27], in non-linear chains [28], in graphene layers [29, 30] and in nanotubes [31].

– *Diffusive to anomalous crossover:* Putting the value $C = \pi/2$ in the expression of $J_{ss}$ in Eq. (68) one rewrites

$$J_{ss} = \frac{T_L - T_R}{4}\frac{1}{\sqrt{N}}\frac{k_o^{3/2}\sqrt{\pi}}{\sqrt{2\gamma}}\left[1 + \sqrt{\frac{N_c}{N}}\,\right], \quad \text{with} \quad N_c = \frac{4}{\pi}\left(\frac{\gamma}{k_o}\right)^3. \tag{69}$$

For fixed $k_o$ and $\gamma$ this expression suggests a crossover from diffusive scaling $\sim \frac{1}{N}$ for $N \ll N_c$ to anomalous scaling $\sim \frac{1}{\sqrt{N}}$ for $N \gg N_c$ as the system size $N$ is increased. We have numerically verified this crossover in fig. 3 where we plot $J_{ss}$ versus $N$ for three choices of the parameters $k_o$ and $\gamma$ such that we have three scenarios of $N_c$ being very small, very large and intermediate. For very large $N_c$ we observe only diffusive

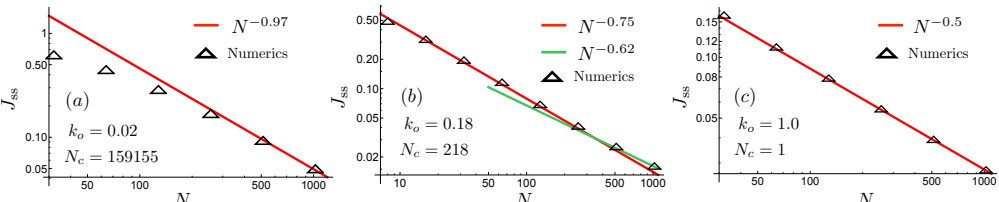

Figure 3: Plot of $J_{ss}$ *vs.* $N$ for $\gamma = 1$ and different values of $k_o$ *i.e.* for different values of $N_c$ as given by Eq. (69). The symbols are obtained by solving the equations of the two-point correlations $\langle \eta_i \eta_j \rangle_{P_{ss}}$ in NESS numerically for different $N$. In (a) we observe diffusive scaling as one has $N \ll N_c = 159155$. In (b), the value of $N_c = 218$. We observe a change in the exponent of the system size scaling from 0.75 to 0.62. In principle, one should observe a crossover from diffusive (exponent 1) to anomalous (exponent 0.5) behaviour. For that one needs to have a sufficiently large $N_c$ so that one has sufficiently large $N$ even for $N < N_c$ to observe the true diffusive scaling and then one should be able to evaluate stationary currents $J_{ss}$ for $N$ much large than $N_c$ to observe the true anomalous scaling. Numerically this is very hard to achieve. Instead, in (c) we choose $k_o = 1.0$ so that one observes only the anomalous scaling because $N_c = 1$.

scaling in fig. 3(a) within the system sizes that were possible to study numerically. Whereas in fig. 3(c) we observe purely anomalous scaling $\sim \frac{1}{\sqrt{N}}$ because $N_c$ is very small. In fig. 3(b), we observe a sort of crossover as manifested by the change in the exponent of the system-size scaling of $J_{ss}$ though not from pure diffusive scaling exponent to the correct anomalous scaling exponent. For that one requires to compute $J_{ss}$ for very large $N$ along with large $N_c$.

– *Anomalous to diffusive crossover:* We end this section by making the following remark: If one considers $k_o$ to be system size dependent as $k_o = N^{-\alpha}$, then for $0 < \alpha < \frac{1}{3}$, one finds, as can be shown following the procedure described in this paper, that the anomalous scaling for the stationary current $J_{ss} \sim \frac{1}{N^{(1+3\alpha)/2}}$ and for $\alpha \geq \frac{1}{3}$ the transport becomes diffusive with $J_{ss} \sim \frac{1}{N}$. This crossover by tuning the strength $k_o$ of the interaction was predicted previously in [32].

## 5  Conclusion

In this paper we have derived macroscopic linearized hydrodynamics for the two conserved quantities present in the HCVE model. Assuming a slowly varying (both in space and time) LE state that is slightly deviated from a underlying GE state, we study the evolution of the average conserved field densities. This is achieved by asking what equations the temperature and the 'pressure' fields (characterizing the LE state) should satisfy to linear order in deviations from their GE values. Approximating the solution of the FP equation by the LE distribution yields linear diffusive hydrodynamics in which one neglects the space-time correlations of the currents corresponding to the conserved fields. In order to include the contributions from such correlations in the linear response regime, we estimate the correction to the local equilibrium distribution in the solution of the relevant FP equation. Such correction naturally produce contributions to the average currents as space-time integrals of the certain current-current correlations. Interestingly, we find that

these current-current correlations involves mainly the non-linear parts of the currents when written in terms of the deviations of the conserved fields from their GE values. To compute such correlations, we invoke fluctuating hydrodynamics equations for the 'volume' field written at a mesoscopic scale. We finally obtain drift-diffusion equation for the 'volume' field and super-diffusion equation for the non-convective part of the energy field to linear order in deviations *i.e.* in the LR regime.

Our calculation, in addition, also provides the diffusive correction to the super-diffusion equation which allows us to study a crossover from diffusive to super-diffusive transport. In particular, our analysis allows us to identify a length scale $N_c$ which depends on the microscopic parameters. Below this length scale, one would observe a diffusive transport and above this length scale the super-diffusive transport sets in. The physical picture is the following: as smaller length scale the conserved quantities dissipate through diffusion. However at larger length scales the correlations among the hydrodynamic currents starts providing dominant channels for transport and as a result one observes a crossover from a diffusive transport to anomalous transport. We have demonstrated this crossover through the system size scaling of the NESS current in the system when connected to two reservoirs of different temperatures at the two ends. Since the HCVE model dynamics is linear (due to harmonic potential), we believe the linearized macroscopic hydrodynamics is exact in the sense there will be no non-linear corrections. However, one can expect to get non-linear corrections both local and non-local in other models such as Fermi-Past-Ulam-Tsingou model. It would be interesting to see how super-diffusion and higher order corrections appear following the formalism presented in this paper. Applying this formalism to non-linear hamiltonian models in open set-up requires to solve NFHD equations in bounded domain with appropriate boundary conditions which, to our knowledge, are not known. It would be interesting to investigate such cases. Often an interesting problem that people consider in systems permitting hydrodynamics description is to observe the evolution from a domain wall initial condition. In systems exhibiting anomalous transport, one should solve the super-diffusion equation for such problems. We believe our result will be useful in such contexts.

# 6 Acknowledgement

The author would like thank Abhishek Dhar for useful comments on the manuscript. A. K. also acknowledge the support of the core research grant no. CRG/2021/002455 and MATRICS grant MTR/2021/000350 from the Science and Engineering Research Board (SERB), Department of Science and Technology, Government of India. Further, A.K. acknowledges support from the Department of Atomic Energy, Government of India, under project no. 19P1112R&D.

# A NFHD for the HCVE model

In this appendix we discuss the NFHD theory for the HCVE model as given in [15]. We start with the conservation equations (22), which in the hydrodynamic limit can be written as

$$
\begin{aligned}
\partial_t h_i(t) &= -2k_o \nabla_i h_i(t) + \gamma \Delta_i^2 h_i(t), \\
\partial_t e_i(t) &= -k_o^2 \nabla_i h_i(t)^2 + \gamma \Delta_i^2 e_i(t).
\end{aligned}
\tag{70}
$$

Writing $h_i(t) = h_0 + \tilde{h}_i(t)$ and $e_i(t) = e_0 + \tilde{e}_i(t)$ and expanding the currents up to second order in the deviations $\tilde{h}_i$ and $\tilde{e}_i$ from the global equilibrium values, one gets

$$\partial_t \tilde{\boldsymbol{u}}_i + \nabla_i \left[ A\tilde{\boldsymbol{u}}_i + \frac{1}{2} \left( \begin{array}{c} \tilde{\boldsymbol{u}}_i^T H^{(h)} \tilde{\boldsymbol{u}}_i \\ \tilde{\boldsymbol{u}}_i^T H^{(e)} \tilde{\boldsymbol{u}}_i \end{array} \right) \right] = 0, \quad \text{with,} \ \tilde{\boldsymbol{u}}_i = \left( \begin{array}{c} \tilde{h}_i \\ \tilde{e}_i \end{array} \right) \tag{71}$$

where for the Harmonic potential $V(\eta) = k_o \eta^2/2$,

$$A = \left( \begin{array}{cc} -2k_o & 0 \\ -k_o^2 h_0 & 0 \end{array} \right), \quad H^{(h)} = \left( \begin{array}{cc} 0 & 0 \\ 0 & 0 \end{array} \right), \quad \text{and} \ \ H^{(e)} = \left( \begin{array}{cc} -2k_o^2 & 0 \\ 0 & 0 \end{array} \right). \tag{72}$$

Adding diffusion and noise terms phenomenologicaslly, one gets the non-linear fluctuating hydrodynamic equations

$$\partial_t \hat{\tilde{\boldsymbol{u}}}_i + \nabla_i \left( A\hat{\tilde{\boldsymbol{u}}}_i + \frac{1}{2} \left( \begin{array}{c} \hat{\tilde{\boldsymbol{u}}}_i^T H^{(h)} \hat{\tilde{\boldsymbol{u}}}_i \\ \hat{\tilde{\boldsymbol{u}}}_i^T H^{(e)} \hat{\tilde{\boldsymbol{u}}}_i \end{array} \right) - \nabla_i \tilde{D} \hat{\tilde{\boldsymbol{u}}}_i + \tilde{B} \tilde{\boldsymbol{\xi}}_i \right) = 0, \tag{73}$$

where $\tilde{D} = \tilde{D}^T > 0$ is the diffusion matrix, and $\xi_i^{(\alpha)}(t)$ for $\alpha = 1, 2$ are white Gaussian noises with zero mean and covariance

$$\langle \xi_i^{(\alpha)}(t) \xi_{i'}^{(\alpha')}(t') \rangle = \delta_{\alpha\alpha'} \delta_{i,i'} \delta(t - t'). \tag{74}$$

The strength of the noise $\tilde{B}\tilde{B}^T$ is related to the diffusion matrix as $\tilde{D}\tilde{C} + \tilde{C}\tilde{D} = \tilde{B}\tilde{B}^T$, where $\tilde{C}$ is the susceptibility matrix given by

$$\langle \hat{\tilde{u}}_\alpha(i,0) \hat{\tilde{u}}_{\alpha'}(i',0) \rangle_{PGE} = \tilde{C}_{\alpha,\alpha'} \delta_{i,i'}. \tag{75}$$

Following [4, 5, 15], one next decomposes the fields $\tilde{\boldsymbol{u}}$ into normal modes $\boldsymbol{\phi}$ using the transformation

$$\hat{\boldsymbol{\phi}} = R\hat{\tilde{\boldsymbol{u}}}, \tag{76}$$

where the matrix $R$ has the properties

$$RAR^{-1} = \left( \begin{array}{cc} -2k_o & 0 \\ 0 & 0 \end{array} \right), \quad \text{and} \ \ RCR^T = 1. \tag{77}$$

For our case $R$ is explicitly given by

$$R = \left( \begin{array}{cc} -\sqrt{k_o \beta_0} & 0 \\ \sqrt{2}\beta_0 \tau_0 & \sqrt{2}\beta_0 \end{array} \right), \tag{78}$$

which implies

$$\hat{\boldsymbol{\phi}}_i = R\hat{\tilde{\boldsymbol{u}}}_i = \left( \begin{array}{c} -\sqrt{k_o \beta_0} \ \hat{\tilde{h}}_i \\ \sqrt{2}\beta_0 \ (\hat{\tilde{e}}_i + \tau_0 \hat{\tilde{h}}_i) \end{array} \right) = \left( \begin{array}{c} -\sqrt{k_o \beta_0} \ \hat{\tilde{h}}_i \\ \sqrt{2}\beta_0 \hat{\tilde{\theta}}_i \end{array} \right), \tag{79}$$

where we have written $\hat{\tilde{\theta}}_i = (\hat{\tilde{e}}_i + \tau_0 \hat{\tilde{h}}_i)$. Note, $\langle \hat{\tilde{\theta}}_i \rangle_{PLE} = \tilde{T}_i/2$ according to Eq. (25). The HD equations (73) can now be written in terms of $\phi_i^{(1)}(t)$ and $\phi_i^{(2)}(t)$ as

$$\partial_t \hat{\phi}_i^{(1)}(t) + \nabla_i \left( -2k_0 \hat{\phi}_i^{(1)}(t) - \nabla_i (D_{11} \hat{\phi}_i^{(1)}(t) + D_{12} \hat{\phi}_i^{(2)}(t)) + (B\boldsymbol{\xi})^{(1)} \right) = 0, \tag{80}$$

$$\partial_t \hat{\phi}_i^{(2)}(t) + \nabla_i \left( -\sqrt{2}k_0 \hat{\phi}_i^{(1)}(t)^2 - \nabla_i (D_{21} \hat{\phi}_i^{(1)}(t) + D_{22} \hat{\phi}_i^{(2)}(t)) + (B\boldsymbol{\xi})^{(2)} \right) = 0, \tag{81}$$

where $D = R\tilde{D}R^{-1} > 0$ and $B = R\tilde{B}$. The matrices $D$ and $B$ now satisfies

$$D + D^T = BB^T. \tag{82}$$

Using Eq. (79), one can express these equations in terms of $\hat{\tilde{h}}$ and $\hat{\tilde{\theta}}$ as

$$\partial_t \hat{\tilde{h}}_i(t) + \nabla_i \left[ -2k_o \hat{\tilde{h}}_i(t) - \nabla_i \left( D_{11} \hat{\tilde{h}}_i(t) - \frac{\sqrt{2\beta_0} D_{12}}{\sqrt{k_o}} \hat{\tilde{\theta}}_i(t) \right) - \frac{(B\boldsymbol{\xi})^{(1)}}{\sqrt{k_o \beta_0}} \right] = 0, \tag{83}$$

$$\partial_t \hat{\tilde{\theta}}_i + \nabla_i \left[ -k_o^2 \hat{\tilde{h}}_i(t)^2 - \nabla_i \left( -\frac{D_{21}}{\sqrt{2k_o \beta_0}} \hat{\tilde{h}}_i(t) + D_{22} \hat{\tilde{\theta}}_i(t) \right) + \frac{(B\boldsymbol{\xi})^{(2)}}{\sqrt{2}\beta_0} \right] = 0, \tag{84}$$

Note, in order for the above equations to be stable one is required to choose $D_{12} = D_{21} = 0$. Hence, the matrix $D$ is a diagonal matrix and consequently by Eq. (82) the matrix $B$ is also diagonal. The fluctuating hydrodynamic equations now look like

$$\partial_t \hat{\tilde{h}}_i(t) + \nabla_i \left[ -2k_o \hat{\tilde{h}}_i(t) - \nabla_i \left( D_{11} \hat{\tilde{h}}_i(t) \right) + \frac{B_{11} \xi^{(1)}}{\sqrt{k_o \beta_0}} \right] = 0, \tag{85}$$

$$\partial_t \hat{\tilde{\theta}}_i + \nabla_i \left[ -k_o^2 \hat{\tilde{h}}_i(t)^2 - \nabla_i \left( D_{22} \hat{\tilde{\theta}}_i(t) \right) + \frac{B_{22} \xi^{(2)}}{\sqrt{2}\beta_0} \right] = 0, \tag{86}$$

Note Eq. (85) is of the same form as the Eq. (38), except the diffusion and the noise terms are introduced phenomenologically and not explicitly known. In the next section, we provide a heuristic derivation of Eq. (38) with explicit dissipation and noise terms.

One of the main prediction of the NFHD theory is the space-time dependence of the correlations

$$S_{\alpha,\alpha'}(i,t) = \langle \hat{g}_\alpha(i,t)\hat{g}_{\alpha'}(0,0) \rangle - \langle \hat{g}_\alpha(i,t) \rangle \langle \hat{g}_{\alpha'}(0,0) \rangle \tag{87}$$

where $\hat{g}_1(i,t) = \hat{h}_i(\vec{\eta}(t))$ and $\hat{g}_2(i,t) = \hat{e}_i(\vec{\eta}(t))$. The space-time correlation of the normal mode fields can be obtained by transforming the matrix $S(i,t)$ as $S^{(\phi)}(i,t) = RS(i,t)R^T$. Since the mode $\phi_i^{(1)}(t)$ gets linearly separated from the non-moving mode $\phi_i^{(1)}(t)$ with time, on sufficiently large space-time scales, the matrix $S^{(\phi)}(i,t)$ is approximately diagonal $i..e.$ $S_{\alpha,\alpha'}^{(\phi)}(i,t) \simeq \delta_{\alpha,\alpha'} \mathcal{F}_\alpha(i/N,t)$. In [15] it was argued that for $V(\eta) = k_o \eta^2/2$, the sound peak $\mathcal{F}_1(i/N,t)$ asymptotically possess diffusive scaling and is described by a Gaussian whereas the heat peak possess anomalous scaling and is described by a $\frac{3}{2}$–Lévy distribution. For other choices of potentials one may get KPZ scaling for the sound mode and $\frac{5}{3}$–Lévy heat mode. A complete classification of the scaling behaviors of the modes for general potential $V(\eta)$ is given in [15].

## B  Heuristic derivation of Eq. (38)

From Eq. (3) and Eq. (5) we rewrite the equation for the 'volume' field explicitly as

$$\partial_t \hat{h}_i = k_o \nabla_i (\hat{h}_i + \tilde{h}_{i-1}) + \gamma \Delta_i \hat{h}_i + \nabla_i \left( \frac{d\mathcal{Z}_{i+1/2}}{dt} \right), \tag{88}$$

$$\text{with} \quad \mathcal{Z}_{i+1/2}(t) = \int_0^t dt' \; [\hat{h}_{i+1}(t') - \hat{h}_i(t')] \left( \frac{dN_{i+1/2}(t')}{dt'} - \gamma \right), \tag{89}$$

where $N_{i+1/2}(t)$ represents the Poisson process describing the exchanges that are happening at the bond $(i, i+1)$ with rate $\gamma$. More precisely, $N_{i+1/2}$ counts the number of exchange events happened till time $t$ on the bond $(i, i+1)$.

Since we are interested in a slowly evolving LE picture in which the conserved fields are slowly varying both over space and time, one can imagine appreciable evolution to occur over time scale that is much larger than the microscopic time scales of the system. Over $\delta t$ time, there are many independent exchange events whose cumulative effect can be obtained following the idea of central limit theorem. Hence we can write

$$\delta \mathcal{Z}_{i+1/2} = \mathcal{Z}_{i+1/2}(t + \delta t) - \mathcal{Z}_{i+1/2}(t), \tag{90}$$

$$= \sum_{k=N_{i+1/2}(t)}^{N_{i+1/2}(t+\delta t)} [\hat{h}_{i+1}(t_k) - \hat{h}_i(t_k)] - \gamma \int_t^{t+\delta t} dt' [\hat{h}_{i+1}(t') - \hat{h}_i(t')] \tag{91}$$

where the set of times $\{t_k\}$ represent the times at which exchanges have occurred in the time interval $t$ to $t + \delta t$. Assuming the field $\hat{h}_i(t)$ is changing slowly, for small $\delta t$ one can approximate the above expression as

$$\delta \mathcal{Z}_{i+1/2} \simeq [\hat{h}_{i+1}(t) - \hat{h}_i(t)] \left(N_{i+1/2}(\delta t) - \gamma \delta t\right), \tag{92}$$

It can be shown that

$$\langle \delta \mathcal{Z}_{i+1/2} \rangle = 0 \tag{93}$$

$$\langle \delta \mathcal{Z}_{i+1/2}^2 \rangle_c = \gamma \delta t \langle (\hat{h}_{i+1}(t) - \hat{h}_i(t))^2 \rangle_{c,P}, \tag{94}$$

$$\simeq \gamma \delta t \langle (\hat{h}_{i+1}(t) - \hat{h}_i(t))^2 \rangle_{c,P_{GE}} + O(\nabla_i \tilde{u}_i), \tag{95}$$

$$= 2\gamma \delta t \left(\frac{T_0}{k_o} + h_0^2\right) + O(\nabla_i \tilde{u}_i), \tag{96}$$

$$\langle \delta \mathcal{Z}_{i+1/2}^m \rangle_c \simeq O(\nabla_i \tilde{u}_i), \quad \text{for} \quad m > 2, \tag{97}$$

where $\langle \hat{o} \rangle_{c,P}$ represents cumulants of $\hat{o}$ evaluated with respect to the distribution $P$. Hence, for $\delta t \to 0$, one can write $\frac{d\mathcal{Z}_{i+1/2}}{dt} = \sqrt{\mathcal{B}} \, \xi_{i+1/2}(t)$ with $\mathcal{B} = 2\gamma \left(\frac{T_0}{k_o} + h_0^2\right)$ at the leading order in deviations. Using this in Eq. (88) gives rise to Eq. (38).

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
