# Peer review of "Super-diffusion and crossover from diffusive to anomalous transport in a one-dimensional system"

_SciPost Physics_

## Round 1 · Referee Report · Anonymous (Referee 1) · 2022-12-10

Strengths

1- Presents an elegant and somewhat elementary derivation of superdiffusive transport in a specific model.

Weaknesses

1- The result should be better framed into the broader literature, to clarify what this new approach adds, compared, e.g., to previous derivations fully within NFHD.

Report

The author analyzes the "Harmonic Chain with Volume Exchange" model, which is known to showcase anomalous heat transport.
In the first part of the paper, the author presents a novel microscopic derivation of heat superdiffusion in a closed chain. In the second part, they shift their focus to an open chain coupled to two thermal reservoirs at different temperatures. Here, adapting the tools introduced in the first part, they show that, at large enough system sizes, the steady heat current and equilibrium profile deviate from the ones expected in systems obeying Fourier law.

A microscopic derivation of superdiffusion equations, even in a simple model, is a challenging problem. In this manuscript, the author manages to make progress in the derivation using only a limited number of physically motivated assumptions and without invoking more "heuristic" arguments up to Eq. (36). Furthermore, the treatment presented here also allows for the study of subleading diffusion corrections.
Therefore I think the manuscript matches the acceptance criteria of SciPost Physics and I would recommend its publication after some minor revisions.

Requested changes

1- The author correctly points out that microscopic derivations of energy superdiffusion have already been obtained for this model. However, he does not comment on these except for saying that the derivation presented here is "simpler". I think it would be necessary to add a more in-depth discussion comparing the previous studies to the current one. This would also be a nice occasion to discuss the advantages of the new derivation. This point is particularly important since part of the derivation relies on some heuristics (Eq. (38)), which is very reminiscent of NLFHD, which can already provide a very concise derivation of some aspect of heat superdiffusion. 2- I think the phrasing of the second line of Eq. (1) is not very clear. In fact, I did not understand what the model was until Eq. (11). It would be helpful if the author could add a line below Eq. (1) to further clarify its meaning. 3- In the introduction, in the paragraph starting with "In the second part of the paper [...]" the author discusses the different behaviour present in a mesoscopic and macroscopic system. By reading the paper, I think the author means that there is a crossover scale $N_C$, such that, if the system size $N<N_C$, transport is well described by Fourier law, while, if $N>N_C$, transport is anomalous. If so, please phrase this more explicitly directly in the introduction. Otherwise could the author please clarify this point? 4- (Optional) The way the argument is phrased, the author starts from the exact equation (15) and only much later the LE part of the evolution is discussed. I think it would make the manuscript more pedagogical to start with Sec. 3.1 and then discuss how to improve on top of it by taking into account the deviations described by $P_d$. 5- The paragraph above Eq. (22) states that the ansatz in Eq. (12) is valid when $P_d$ is small. Could the author please clarify what this means, as precisely as possible after Eq. (12). Which exact assumptions are needed for the derivation to go through? 6- Figure (2), panel (b), shows a scaling collapse trying to give numerical evidence for superdiffusive scaling. The data is however not very convincing at the moment, as the time goes from s=70 to s=80. I think it's very hard to determine the dynamical exponent from such a small time window. Could the author please justify why such a small time window is used? If the range cannot be extended (for reasons that should be explained), one way to make the numerical data look more convincing would be to present in an inset or an appendix an attempt at collapsing the same data with different exponents near 2/3 (including 1/2) and show that their agreement is much worse. 7- Many acronyms are used in the manuscript, and some of them appear very few times, e.g. LR appears only 5 times. It would help clarity to expand the acronyms that are not used often.

  • validity: high
  • significance: good
  • originality: high
  • clarity: high
  • formatting: excellent
  • grammar: perfect

Author:  Anupam Kundu  on 2023-02-13  [id 3350]

(in reply to Report 1 on 2022-12-10)
Category:
answer to question

Please see the attached report.

Attachment:

report-1.pdf

---

## Round 1 · Referee Report · Anonymous (Referee 2) · 2022-12-19

Strengths

  • very interesting calculation of the dynamics of the model, without assuming NLFH

Weaknesses

  • could be written slightly more clearly, also it is unclear what this approach gives more than NLFH

Report

The author consider a system with two hydrodynamics modes in 1D, volume and energy, where non-linear fluctuating hydrodynamics can be employed to show that volume is diffusive while energy is super diffusive. The author takes another approach, he studies the dynamics of a small perturbation on top of the local steady state and shows the same phenomenology. It is an interesting calculation and it deserves publication. My remarks are:

I would require the author to clarify in the abstract that what he found is an agreement with NLFH and also to clarify why he thinks that NLFH cannot provide the crossover from diffusion to superdiffusion.

Could the author write a clear and full derivation of eq 15? It should be clear that the solution for P_d is in the linear response.

Requested changes

see above

  • validity: good
  • significance: high
  • originality: high
  • clarity: good
  • formatting: excellent
  • grammar: excellent

Author:  Anupam Kundu  on 2023-02-13  [id 3349]

(in reply to Report 2 on 2022-12-19)
Category:
answer to question

Please see the attached file.

Attachment:

report-2.pdf

---

## Round 1 · Referee Report · Anonymous (Referee 3) · 2023-3-23

Strengths

  • a nice and rather precise derivation of linearised hydrodynamic equations, showing diffusion and super-diffusion. I find the calculation, in this specific model, very illuminating and helpful to understand linearied hydrodynamics more generally.

Weaknesses

  • notation a bit heavy, logical flow at places can be improved.

Report

In this paper, the author provides a nice derivation of linearised hydrodynamic equations for a particular microscopic, stochastic many-body model. The model is simple enough that it allows for exact calculations of the hydrodynamic expansion. The resulting hydrodynamic coefficients, which appear as space-time integrals of current-current correlators (as is generally expected), are evaluated using the approximation of fluctuating hydrodynamics. The linearised hydrodynamic equations display interesting physics, especially super-diffusive behaviour, which is encoded within a non-local hydrodynamic coefficient.

Overall I find the calculation very illuminating, and the results interesting. The derivations are at times hard to follow because the symbols used are a bit heavy, and there are some very small gaps in the logic (see below), but overall this is a very nice paper that I enjoyed reading.

I fully recommend publication in Scipost, once the comments below have been addressed.

  • On the general side, I have only one technical question: it is sometimes said that one may be able to write terms in the linearised hydrodynamic equations with non-trivial scaling using fractional derivatives - which are indeed non-local objects. Is it possible to interpret the results here, e.g. the first term in the brackets on the r.h.s. of eq 53, in terms of fractional derivatives?

  • Here are comments that I had while reading the paper; some small, some would require improved discussion / re-ordering.

$k_0$ should be $k_o$ in caption of Figure 1

In eq 8 or around: write that in fact $\langle \hat h_i\rangle = h_0$ and $\langle \hat e_i\rangle = e_0$for all i, to make it clearer.

Eq 17: why put $(\cdots)_{LE}$ in the time derivatives on the right-hand side? These are just derivatives of the functions $T_i(t)$ and $\tau_i(t)$ defined in eq 13. There is a sentence on page 7 that gives an explanation, but I think the logic here is a bit unclear. In fact, eqs 16-17 are a formal solution for any choice of time dependence for $P_{LE}(\vec\eta,t)$. But that the author really wants to do is to choose $P_{LE}(\vec\eta,t)$ to solve the hydrodynamic equations. This should be explained already, either just after eq 14, before the formal solution is given, or just after that. It should be explained that the general solution is valid for any choice of time dependence for $P_{LE}(\vec\eta,t)$, but that one gets good control over the solution in the large-scale (slowly varying) limit, by assuming $P_d$ is small, which will imply that $P_{LE}(\vec\eta,t)$ solves the hydrodynamic equation, and by looking at corrections to this. Also concerning the paragraph above eq 22: again eq 12 is in fact not an ansatz. It is just a way of expressing the solution. $P_d(\vec \eta,t)$ is any function of $\vec\eta,t$, hence this way of writing the solution is always valid. It becomes an ansatz if we assume that $P_d$ is small, as is done in sec 3.1.

Around eq 24: it is not clear how this scaling discussion fits the general discussion up to there and just following it. It looks like nowhere the slowly varying assumption is being used up to there. It is also not used explicitly in section 3.1. Naturally, the assumption will be important in order to justify that $P_d$ is indeed small if we choose $P_{LE}$ to solve the hydrodynamic equation; its smallness will be controlled by $\epsilon$. I would suggest to put the hydrodynamic scaling discussion at the end of sec 3.1 instead, where this should be explained.

After eq 30: the y’s are referred to as currents. They are not exactly the currents, as the stochastic terms are missing. What are they exactly? Perhaps this should be discussed already around eq 18, where they appear first.

After eq 32: I find the discussion of linear / non-linear terms a bit confusing. Here, we want to keep linear terms in the deviations of averages from GE, but we keep all terms in the deviations of fundamental fluctuating quantities (those that have a hat). So we keep the product $\hat{\tilde h_i}\hat{\tilde h_{i+1}}$. This is an important point!

Around eq 34: here the order of limit is incorrect. The limit $N\to\infty$ should be taken first. This is corrected in sec 3.2.1, however it should be expressed correctly already here. We are interested in the evolution at macroscopic scale, but this means not only time, but sizes also are macroscopic.

Eq 35,36: this looks like the standard Kubo formula for for the diffusive correction to the Euler hydrodynamic approximation of the current. However, the standard formula involves the current-current correlation, while here it is the operator $\hat{\mathcal T}^{(e)}_{\ell,\ell+1}$. The two-point function is later referred to as “current-current”. But the operator $\hat{\mathcal T}^{(e)}_{\ell,\ell+1}$ is not exactly the microscopic current; terms are missing. Could the author explain a bit more here? Also, in eq 34 the position of $T_\ell(t)$ is summed over; it is not the position $i$. This is also different from the standard Kubo formula. Shouldn’t we recover the standard formula here? More explanations would be useful.

After eq 40 “where $\delta_{i,j}$ is the Kronecker delta” seems to be misplaced.

Beginning of sec 3.3 (and also before, where the scaling is introduced in eq 24, and also on page 15): the $i\to i\epsilon$ and $t\to \bar\epsilon t$ scaling is not accurate for the pressure field, because in the actual transformation both $\epsilon t$ and $\bar\epsilon t$ appear (because there is a ballistic front) - re-write the scaling to make it consistent.

Paragraph after eq 54: $x\sqrt{s}$ seems to have a missing $\sim$ in-between.

Fig 2: the diffusive and super-diffusive scaling are indeed very apparent. But would it be possible to compare with the exact hydrodynamic formula? It should be possible to evaluate these time-independent profiles of rescaled temperatures from the hydro formulae 49 (simple) and especially 53 (more interesting), the latter at least as some kind of integral equations that can be solved numerically. This way we would see the exact constants appearing - this is a no-parameter fit.

Pages 16-17: some of the remarks there apply to the case without boundaries as well. It may be clearer to make a separate “discussion” or “remark” section, to put things together in a clear discussion.

Requested changes

See the report!

---

## Editorial Decision

resubmitted